# A national survey integrating clinical, laboratory, and WASH data to determine the typology of trachoma in Nauru

Kathleen D. Lynch[1]*, Sue Chen Apadinuwe[2], Stephen B. Lambert[1,3], Tessa Hillgrove[4], Mitchell Starr[5], Beth Catlett[5], Robert S. Ware[6], Anasaini Cama[4], Sara Webster[4], Emma M. Harding-Esch[7], Ana Bakhtiari[8], Robert Butcher[7], Philip Cunningham[5], Diana Martin[9], Sarah Gwyn[9], Anthony W. Solomon[10], Chandalene Garabwan[2], John M. Kaldor[11], Susana Vaz Nery[11]

1 UQ Centre for Clinical Research, University of Queensland, Brisbane, Australia, 2 Ministry of Health and Medical Services, Denig, Republic of Nauru, 3 National Centre for Immunisation Research and Surveillance, Westmead, New South Wales, Australia, 4 The Fred Hollows Foundation, Melbourne, Australia, 5 NSW State Reference Laboratory for HIV, St Vincent's Centre for Applied Medical Research, St Vincent's Hospital, Sydney, Australia, 6 Menzies Health Institute Queensland and School of Medicine, Griffith University, Brisbane, Australia, 7 Clinical Research Department, London School of Hygiene & Tropical Medicine, London, United Kingdom, 8 International Trachoma Initiative, Task Force for Global Health, Decatur, Georgia, United States of America, 9 Division of Parasitic Diseases and Malaria, Centers for Disease Control and Prevention, Atlanta, Georgia, United States of America, 10 Department of Control of Neglected Tropical Diseases, World Health Organization, Geneva, Switzerland, 11 The Kirby Institute, University of New South Wales, Sydney, Australia

* kathleen.lynch1@uqconnect.edu.au

## Abstract

### Background

The epidemiology of trachoma in several Pacific Islands differs from other endemic settings, in that there is a high prevalence of clinical signs of trachoma, particularly trachomatous inflammation—follicular (TF), but few cases of trichiasis and limited evidence of ocular chlamydial infection. This so-called "Pacific enigma" has led to uncertainty regarding the appropriate public health response. In 2019 alongside Nauru's national trachoma population survey, we performed bacteriological and serological assessments of children to better understand the typology of trachoma and to determine whether there is a need for trachoma interventions.

### Methods

We used two-stage cluster sampling, examining residents aged ≥1 year and collecting household-level water, sanitation, and hygiene (WASH) variables. Children aged 1–9 years provided conjunctival swabs and finger-prick dried blood spots to investigate the presence of *Chlamydia trachomatis* nucleic acid and anti-Pgp3 antibodies, respectively.

### Principal Findings

In 818 participants aged 1–9 years, the age-adjusted TF prevalence was 21.8% (95% CI 15.2–26.2%); ocular *C. trachomatis* prevalence was 34.5% (95% CI 30.6–38.9), and anti-

WHO logo is not permitted. This notice should be preserved along with the article's original URL.

**Data Availability Statement:** All relevant data are within the manuscript and its Supporting Information files

**Funding:** This research was supported by The Fred Hollows Foundation using UK aid funding (primary grant held by Sightsavers International) this included salary support for AC and SVN. SC and CG received funds from The Fred Hollows Foundation for their role in implementing the trachoma survey. Core Tropical Data funding was provided by the International Trachoma Initiative; Sightsavers; and RTI International through the United States Agency for International Development (USAID) Act to End NTDs East program. AWS is a staff member of the World Health Organization. RB's salary is funded by The Fred Hollows Foundation (ref 1954-0). Staff from The Fred Hollows Foundation participated in the development of the study design, study preparation, staff training, and commenting on the draft manuscript.

**Competing interests:** I have read the journal's policy and the authors of this manuscript have the following competing interests: AB is employed by the International Trachoma Initiative at The Task Force for Global Health, which receives an operating budget and research funds from Pfizer Inc., the manufacturers of Zithromax (azithromycin). EMHE receives salary support from the International Trachoma Initiative.

Pgp3 antibody prevalence was 32.1% (95% CI 28.4%–36.3%). The age- and gender-adjusted prevalence of trichiasis in ≥15-year-olds was 0.3% (95% CI 0.00–0.85), but no individual with trichiasis had trachomatous scarring (TS). Multivariable analysis showed an association between age and both TF (OR per year of age 1.3 [95% CI 1.2–1.4]) and anti-Pgp3 positivity (OR 1.2 [95% CI 1.2–1.3]). There were high rates of access to water and sanitation and no WASH variable was associated with the presence of TF.

## Conclusions

TF, nucleic acid, and age-specific antibody prevalence collectively indicate that high levels of *C. trachomatis* transmission among children present a high risk of ocular damage due to trachoma. The absence of trichiasis with trachomatous scarring suggest a relatively recent increase in transmission intensity.

## Author summary

In contrast to several neighbouring Pacific Island nations, Nauruan children are heavily affected by active trachoma and the cause is ocular infection with *C. trachomatis*. Comprehensive public health intervention to control trachoma in Nauru is required. The use of laboratory markers for current and previous *C. trachomatis* infection should be considered in baseline trachoma prevalence surveys as we approach global elimination of trachoma, and in settings with inconsistent findings during previous screening exercises.

## Introduction

Trachoma is the leading infectious cause of blindness worldwide and was targeted in 1996 for elimination as a public health problem by the year 2020 [1,2]. The road map for neglected tropical diseases 2021–2030 has revised this target to 2030 [3]. Active trachoma is an inflammatory response to ocular infection with *Chlamydia trachomatis*. In trachoma-endemic areas, active trachoma, defined as trachomatous inflammation—follicular (TF) or trachomatous inflammation—intense (TI) in one or both eyes, is common among young children [4]. Repeated episodes of active trachoma cause chronic inflammation which can gradually lead to scarring of the inside of the eyelid (trachomatous scarring [TS]). Progressive conjunctival scarring can distort the eyelid margin and cause the eyelashes to turn inwards, touch the eyeball (trachomatous trichiasis [TT]), and painfully abrade the cornea [4]. Without intervention, the constant abrasion can lead to corneal opacity (CO) and irreversible blindness [5]. It is important to note that while there are other causes of trichiasis, TT is the result of progressive trachomatous scarring [6]. Trachoma elimination programmes employ the World Health Organization (WHO)-endorsed "SAFE" strategy: surgery for trichiasis, antibiotic treatment to clear infection, and facial cleanliness and environmental improvement to reduce transmission of ocular *C. trachomatis* [7]. These "AFE" interventions are recommended in evaluation units (EUs) where the prevalence of TF is ≥5% in children aged 1–9 years, and "S" is recommended where the prevalence of trichiasis is ≥0.2% in those aged ≥15 years (EUs are generally equivalent to a districts, which for trachoma elimination purposes WHO defines as "the normal administrative unit for health care management, consisting of a population unit between 100,000–250,000 persons") [7,8].

Population-based surveys are undertaken to determine whether prevalence thresholds are exceeded in EUs where there has been recent evidence of trachoma. In 2007, a rapid assessment in Nauru found high levels of TF in children (20.7%–33.0%), but no trichiasis in older residents [9,10]. Rapid assessments use a "quick-and-epidemiologically-dirty" approach of convenience sampling for recruiting participants and are specifically designed to overestimate the percentage of children with TF. Whilst results could not be used as a prevalence estimate for programmatic planning, the findings did suggest that active trachoma may be a public health problem and provided justification for further investigation [11]. Subsequent population-based surveys in the neighbouring countries of Papua New Guinea (PNG), Solomon Islands, and Vanuatu had similar findings [12–15]. Additionally, there was little indication in those countries of current *C. trachomatis* infection in children, based on polymerase chain reaction (PCR) detection on conjunctival swabs, or past infection based on antibodies to the *C. trachomatis* antigen Pgp3 [12,15–17]. It was therefore hypothesised that in these three Melanesian nations, as-yet undetermined aetiologies may be responsible for much of the clinical syndrome that is phenotypically indistinguishable from TF, a phenomenon referred to as the "Pacific enigma" [16].

In July 2019, the Nauru Ministry of Health and Medical Services (MHMS) led a population-based trachoma survey using standard Tropical Data methods to assess the prevalence of clinical signs of trachoma and to collect data on household-level water, sanitation and hygiene (WASH) variables [18]. Given the findings of Nauru's 2007 rapid assessment and the observations from neighbouring countries, the survey added a research component testing for evidence of current or past chlamydial infection in 1–9-year-olds. While a few previous studies have collected clinical, bacteriological, serological, and WASH variables, this study is the first nationwide trachoma survey in which all four types of data were generated simultaneously.

## Methods

### Ethics statement

Ethics approval for Tropical Data (https://www.tropicaldata.org/) support for the prevalence survey was provided by the London School of Hygiene & Tropical Medicine Ethics Committee (reference 16105). Ethics approval for the prevalence survey, collection, and testing of samples was granted by the University of New South Wales Human Research Ethics Committee (reference HC190442) and the MHMS, Nauru (17/06/2019). Verbal consent was obtained for clinical examination. A parent or guardian provided written consent for each child to provide biological samples. Direct contact with survey participants was undertaken entirely by MHMS staff. United States Centers for Disease Control and Prevention (CDC) staff did not interact with survey participants or have access to any identifying information. This report conforms with STrengthening the Reporting of OBservational studies in Epidemiology (STROBE) guidelines [19].

### Trachoma survey design

The population-based prevalence survey was undertaken by the Nauru Ministry of Health with the support of The Fred Hollows Foundation using standard Tropical Data methods. Tropical Data builds on the systems and methods of the Global Trachoma Mapping Project [20,21]. The survey was restricted to Nauruan citizens, who comprise most of the population, and excluded foreign citizens working in Nauru and individuals seeking refugee status in Australia. We recruited survey participants using two-stage cluster sampling following standard Tropical Data methods, conforming to WHO recommendations [22,18]. Nauru's estimated resident citizen population of 9,600 was surveyed as a single EU [2,10]. The sample size was

calculated based on a targeted level of precision for estimation of a proportion, via the formula n = DEFF x p(1-p)/(2 x d/(1.96 x 2)$^2$), with correction for a small, finite population [23]. Our estimated sample size requirement for 1–9-year-olds was 724, assuming a TF prevalence of 10% and a design effect of 2·65 to achieve a 95% confidence interval (CI) of width 6% [23]. It was increased by 25% to account for non-response, resulting in 905 children aged 1–9 years to be enumerated. In the first stage, 20 clusters with approximately equal population sizes were created using the fourteen administrative districts in Nauru plus six further subdivisions drawn within the three most populated districts [18]. In the second stage, 23 households were randomly sampled from each cluster using a pre-existing household list held by Nauru MHMS. Assuming a mean of two children aged 1–9 years per household, this sampling frame was expected to yield 920 resident 1–9-year-olds.

Teams conducted the survey in house-to-house visits in July 2019. Each team consisted of a minimum of four individuals: a team leader, certified grader, certified recorder, and a dried blood spot (DBS) specimen collector.

## Household-level data collection

For consenting households, location (using GPS) was recorded and household-level WASH variables obtained from the self-nominated head of the household using standard Tropical Data methods described previously [22]. As per these methods, ten standard questions were asked about household level WASH access; data on handwashing facilities and access to toilet/latrine (if shared) were collected through head of household report [22]. Sanitation facilities were categorised as "improved" or "unimproved", using definitions set by the WHO–UNICEF Joint Monitoring Programme (JMP) for Water Supply and Sanitation [24]. Handwashing facilities were only recorded if there was a latrine in the household, and were categorised as either "available with water and with soap", "available with water but without soap", or "no functioning handwashing facility available" [22]. All six recorders successfully passed the Tropical Data prevalence survey training programme which involves specific training and assessment of recorders [22]. All household- and individual-level data, including verbal consent, were captured electronically in real time using a purpose-built Secure Data Kit-based Android phone application, following standard Tropical Data procedures [22].

## Clinical examination

Prior to conducting the survey, graders and recorders were required to pass internationally standardised training using the Tropical Data system [22]. In the week prior to the survey, field graders undertook a four-day training programme conducted by certified Tropical Data trainers using standardised training material. During the classroom session on examination techniques, trainees were taught how to recognise trichiasis (upper and lower eyelid) and TS. Consistent with the Tropical Data collection method, trainees were taught to recognise and collect data on only moderate or severe scarring as per the WHO simplified grading system definition of TS.

All residents of selected households ≥1 year of age for whom consent was given were examined for trachoma according to the WHO simplified trachoma grading scheme, using binocular 2.5× magnifying loupes and sunlight or a torch for illumination [5]. Participants found to have trichiasis (at least one eyelash touching the eyeball, or evidence of recent epilation of in-turned eyelashes) were further examined for TS and asked whether they had previously been offered surgery for trichiasis or advice to epilate, then referred to the national hospital for management. Participants with active trachoma were provided with two tubes of 1% tetracycline eye ointment.

## Collection of conjunctival specimens and dried blood spots

The research component was undertaken alongside the national survey to collect information on laboratory correlates of clinical findings. Conjunctival specimens were collected by certified graders with assistance for specimen collection provided by the team leader and DBS specimen collector. The assistants were responsible for tube holding and specimen labelling. In the week prior to the survey, the team undertook training in conjunctival and DBS sample collection. This training incorporated infection control procedures to minimise the risk of specimen contamination in the field. For the first three days of fieldwork, specimen collection was supervised by a senior investigator.

Consent for participation in the research component of this work was sought from carers for individuals aged 1–9 years of age who were participating in the national survey. Conjunctival specimens for PCR testing were collected from the left (second-examined) eye from consented 1–9-year-olds following clinical examination. Using the Media Dual Swab Sample Packet (P/N: 05170516190, [Cobas, Roche Diagnostics, New Jersey, USA]), a sterile woven swab was passed three times (with a 120-degree turn between each pass) over the everted upper tarsal conjunctiva and then placed immediately into a Cobas PCR media tube without further contact. Control procedures were implemented to avoid cross contamination of specimens in the field. We took care to ensure the swab head was only in contact with an individual's conjunctiva, and graders cleaned their hands with alcohol-based hand gel between each subject.

DBS for serological testing were collected via finger prick to absorb up to five spots (estimated 50 μL of whole blood per spot) onto Whatman 903 Protein Saver Cards (ThermoFischer Scientific Australia Pty Ltd, Victoria). The staff member collecting the DBS wore gloves, which were changed, and hand hygiene performed, after each participant. Swabs and DBS were carried with the teams for up to 8 hours and stored at room temperature (RT) at the end of each day. DBS cards were air-dried overnight, then each packed into an individual Whatman foil barrier bag (Bio-Strategy, VIC, Australia) containing two 1 g silica desiccant sachets (Desicco Pty Ltd NSW Australia). Swabs and DBS were stored from one to five weeks in Nauru at RT before being transported at ambient temperature to St Vincent's Centre for Applied Medical Research, Sydney, Australia and stored at 2–8˚C and −20˚C, respectively, until laboratory testing. Detailed methods for PCR and serological testing are provided (see S1 Text).

## PCR testing for ocular *Chlamydia trachomatis*

DNA was extracted from the swabs and tested for *C. trachomatis* via real-time PCR (Cobas 6800, Roche Diagnostics), using the Cobas *C. trachomatis* / *Neisseria gonorrhoeae* (CT/NG) dual assay; however, only the *C. trachomatis* results were evaluated as part of the study. Qualitative results were classified according to the manufacturer's instructions.

We implemented rigorous control measures to prevent and identify contamination in the laboratory. There were separate and physically distinct areas in the laboratory designated for sample preparation, DNA extraction and PCR amplification. Each sample was uncapped in a biological safety hood class II, placed on board racks with the collection swab remaining inside the tube, then directly placed onboard the Roche COBAS 6800 instrument. Gloves were changed in between handling test reagents and samples. Testing was performed according to standard operating procedures (SOPs) and manufacturer's Instructions for Use. The assay reagents include uracil-n-glycoslyase, which prevents carry over contamination onboard the instrument. Any contaminating amplicon from previous PCR runs are eliminated by the AmpErase enzyme, which is included in the PCR master mix, during the first thermal cycling step.

### Serological testing for anti-Pgp3 antibodies

**Enzyme-linked immunosorbent assay.** Samples eluted from DBS were tested by an enzyme-linked immunosorbent assay (ELISA) to detect the anti-*C. trachomatis* antibody anti-Pgp3 [25]. DBS samples and dried serum spots (DSS) were tested in duplicate. A blank spot was punched after each patient spot to reduce the risk of carry over contamination. The quality control criteria required to pass each testing plate followed CDC Instructions for Use. DSS positive for anti-Pgp3 antibodies across a range of responses, as well as negative human serum (NHS), were run on each plate [26]. Optical density was measured at 450nm on a Sunrise plate reader (Tecan Group Ltd, Switzerland). The average result from 6 blank wells was subtracted from each averaged absorbance value, and the result normalised against the mid-range positive control to account for plate-to-plate variation [25]. Plates were acceptable if results for 3 out of the 4 positive controls and the NHS control fell within pre-determined ranges (established during assay validation) and the positive controls were at least 50 times higher than background. Initially, 4 out of 21 plates (19%) did not meet quality control criteria provided by the CDC. The samples in these plates were re-punched, re-eluted and re-tested. Each repeat plate then passed quality control criteria. A finite mixture model was used to classify the samples as seropositive or seronegative based on normalized absorbance values. The threshold for seropositivity was determined to be 0.072 by taking the mean of the Gaussian distribution of the seronegative population plus four standard deviations above the seronegative population. Detailed methods for serological testing are provided (S1 Text).

Laboratory staff were blinded to all clinical results and samples were de-identified. The serological and molecular work was performed by two different groups in the laboratory and these groups were kept blinded to each other's results.

### Data analysis

Results of the laboratory testing were linked to the clinical results of the national survey and analysed together. TF prevalence was adjusted for age, and trichiasis prevalence for age and gender using standard methods for trachoma surveys, as previously described [20]. Prevalence (by PCR) and serology (by ELISA) were also adjusted for age. Data from Nauru's 2011 census report were used as the reference dataset for the purposes of undertaking these adjustments [27]. Outcome variables were TF, *C. trachomatis* PCR positivity, and anti-Pgp3 seropositivity by ELISA. Logistic regression analyses were used to assess relationships between these outcomes and age, gender, and household-level WASH variables. First univariable analyses were undertaken, and then multivariable analyses with age, gender, time to get drinking water, time to get washing water, and latrine classification included as covariables. A post-hoc power calculation was undertaken to investigate the detectable alternative for the association between a two-category WASH variable and TF. For 818 children, assuming the variable had an 80:20 split (e.g. 654 in one category vs 164 in the other) and there was 25% TF in the reference WASH group, we had 80% power to detect a statistically significant difference if the percentage of TF in the comparator WASH group was 36% or greater. Cluster-robust variance estimates were calculated using sandwich estimators to account for the sampling scheme. Effect estimates are presented as crude odds ratios (ORs) and adjusted odds ratios (aORs) with 95% confidence intervals (CIs). Average annual seroconversion rate and associated 95% CI were calculated using a generalised linear model with binomial family and identity link function. Statistical analyses were carried out using R (R Project for Statistical Computing, Vienna, Austria) and Stata v16.0 (StataCorp, College Station, TX, USA).

## Results

A total of 459 households were visited and 2,515 people (53.3% female) examined across 20 survey clusters. 98 people declined to participate, and one adult could not be examined due to an ocular abnormality (Fig 1). Household-level WASH data were obtained for all households. Of 818 participating children aged 1–9 years, 96.2% (n = 787) resided in a household with access to an improved source of drinking water, 95.7% (n = 783) with access to an improved source of washing water, and 88.9% (n = 727) with access to an improved latrine.

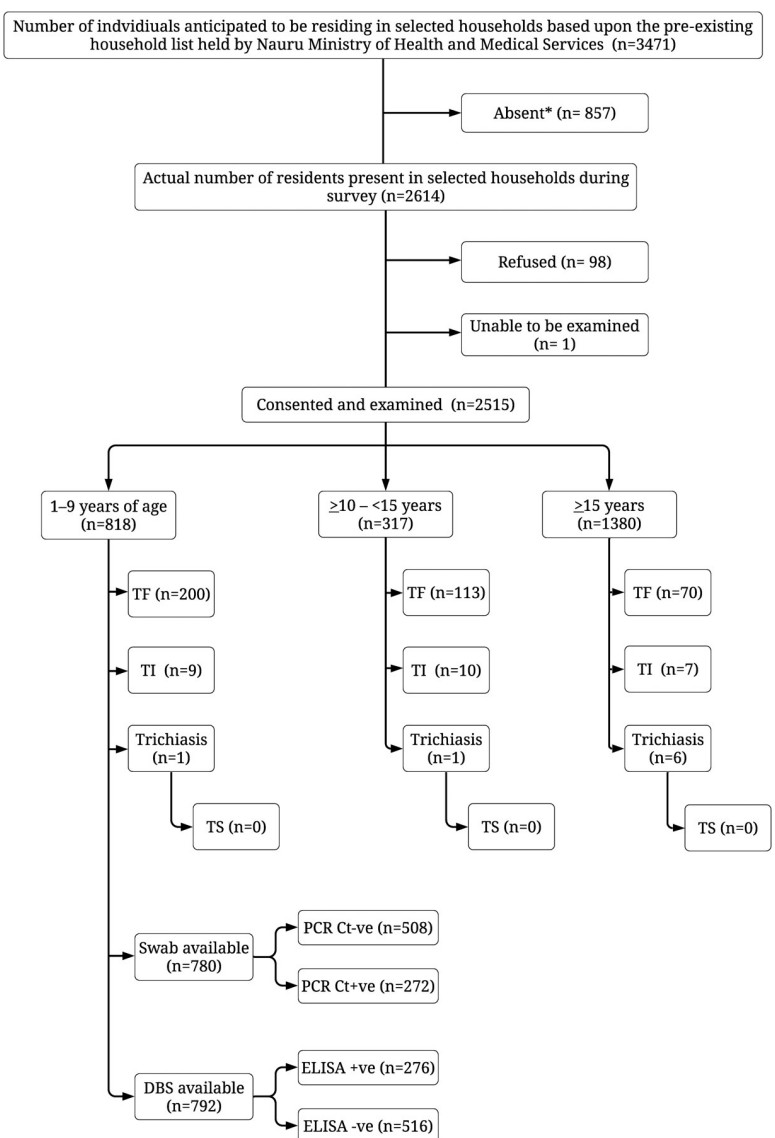

**Fig 1. Recruitment and results flow diagram, Nauru, July 2019.** Abbreviations: TF, trachomatous inflammation—follicular; TI, trachomatous inflammation—intense; TS, trachomatous scarring; PCR Ct, polymerase chain reaction *Chlamydia trachomatis*; ELISA, enzyme-linked immunosorbent assay; +ve, positive; -ve, negative. *This is a slight underestimation due to inconsistent recording of absenteeism in the first week of the survey. During this first week, 16% (124/764) of people were recorded as absent vs 27% (733/2707) across the remaining 3 weeks of the survey. Extrapolating the average level of absenteeism over the last three weeks to the first week, we expect an additional 82 individuals were absent.

### Examination findings

Of the 818 children aged 1–9 years, 200 (24.4%) had TF and 9 (1.1%) had TI (Fig 1). The age-adjusted prevalence of TF in children was 21.8% (95% CI 15.2–26.2). Of 317 children aged 10–14 years, 35.6% (113) had TF. The prevalence of TF was 3% for 1-year-olds and increased with age to 38% in 9-year-olds (Fig 2), with a 1.28-fold increase in adjusted odds of having TF for each additional year of age within that age range (Table 1; 95% CI 1.18–1.40). Gender did not affect the odds of TF (Table 1; aOR 0.73, [95% CI 0.49–1.09]). TI prevalence increased with age, from 0.3% (1/346) of children <4 years, 1.7% (8/472) of children aged 5–9-years, to 35.6% (113/317) of those aged 10–14 years.

There were six (0.4%) cases of trichiasis identified among 1,380 participants aged ≥15 years. All six cases were unilateral. Five of the cases were in individuals aged 15–39 years (60% male), and one in a female participant aged ≥40 years. There were two cases in children, a 9-year-old [male] and a 13-year-old [female] (Fig 1). Both cases were unilateral. The age- and gender-adjusted prevalence of trichiasis was 0.3% (95% CI 0.00–0.85). It was reported specifically by the graders that no individual with trichiasis had TS (Fig 1). In individuals aged ≥15 years, 5.1% (70/1380) had TF and 0.5% (7/1380) had TI.

### Ocular *C. trachomatis* infection

PCR data on infection were missing for 4.6% of participants comprising 8 who refused, 14 had no swab collected and 16 for whom labelling errors prevented matching of PCR and clinical data. Of the 780 children for whom a swab was available, 272 (34.9%) were PCR-positive for *C. trachomatis*. The age-adjusted infection prevalence was 34.5% (95% CI 30.6–38.9). PCR

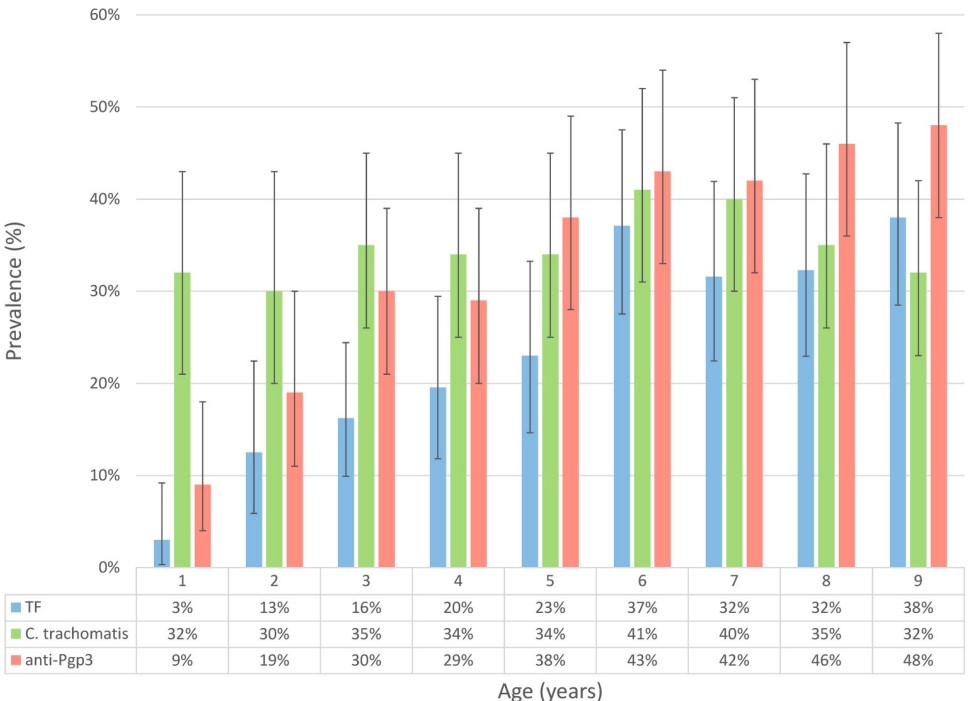

| Age (years) | 1 | 2 | 3 | 4 | 5 | 6 | 7 | 8 | 9 |
|---|---|---|---|---|---|---|---|---|---|
| TF | 3% | 13% | 16% | 20% | 23% | 37% | 32% | 32% | 38% |
| C. trachomatis | 32% | 30% | 35% | 34% | 34% | 41% | 40% | 35% | 32% |
| anti-Pgp3 | 9% | 19% | 30% | 29% | 38% | 43% | 42% | 46% | 48% |

**Fig 2. Prevalence of trachomatous inflammation—follicular, ocular *Chlamydia trachomatis* detection, and anti-Pgp3 antibodies in examined children aged 1–9 years in Nauru, July 2019.** Abbreviations: TF, trachomatous inflammation—follicular. Error bars showing 95% confidence intervals around survey prevalence estimate for each outcome.

**Table 1. Factors associated with trachomatous inflammation—follicular (TF) in children aged 1–9 years, Nauru, July 2019 (n = 818).**

| Variable | n | TF[+ve] n (%) | OR (95%CI); p-value | aOR (95%CI); p-value[a] |
|---|---|---|---|---|
| **Age, increase per year** | | | | |
| 1–9 years | 818 | 200 (24.5) | 1.27 (1.17–1.37); <0.001 | **1.28 (1.18–1.40); <0.001** |
| **Gender** | | | | |
| Male | 426 | 112 (26.3) | 1.0 (reference) | |
| Female | 392 | 88 (22.5) | 0.81 (0.57–1.16); 0.25 | 0.73 (0.49–1.09); 0.13 |
| **Household source of water used for drinking** | | | | |
| Improved | 787 | 200 (25.4) | 1.0 (reference) | 1.0 (reference) |
| Unimproved | 24 | 0 | no events | no events |
| Other | 7 | 0 | no events | no events |
| **Time to get drinking water** | | | | |
| Water source in the yard | 552 | 161 (29.2) | 1.0 (reference) | 1.0 (reference) |
| Travel required | 266 | 39 (14.7) | 0.42 (0.19–0.90); **0.03** | 0.47 (0.07–3.10); 0.43 |
| **Household source of water used for washing[b]** | | | | |
| Improved | 783 | 199 (25.4) | 1.0 (reference) | 1.0 (reference) |
| Unimproved | 30 | 1 (3.3) | 0.10 (0.01–1.46); 0.09 | 0.16 (0.01–1.20); 0.15 |
| **Time to get washing water** | | | | |
| Water source in the yard | 401 | 124 (30.9) | 1.0 (reference) | 1.0 (reference) |
| All face washing done at the source | 152 | 36 (23.7) | 0.69 (0.44–1.09); 0.12 | 0.76 (0.46–1.26); 0.29 |
| Travel required | 265 | 40 (15.1) | 0.39 (0.18–0.87); **0.02** | 0.80 (0.12–5.28); 0.82 |
| **Where do adults in the household usually defecate?** | | | | |
| Private latrine | 788 | 189 (24.0) | 1.0 (reference) | 1.0 (reference) |
| Other | 30 | 11 (36.7) | 1.83 (0.88–3.81); 0.10 | 1.68 (0.88–3.23); 0.12 |
| **Household latrine** | | | | |
| Improved | 727 | 188 (25.9) | 1.0 (reference) | 1.0 (reference) |
| Unimproved | 91 | 12 (13.2) | 0.44 (0.12–1.55); 0.20 | 0.68 (0.20–2.31); 0.54 |
| **Is there a functioning handwashing facility available?** | | | | |
| Handwashing available with water and with soap | 713 | 173 (24.26) | 1.0 (reference) | 1.0 (reference) |
| Handwashing available with water but without soap | 34 | 6 (17.65) | 0.68 (0.32–1.39); 0.28 | 0.57 (0.22–1.48); 0.25 |
| No functioning handwashing facility available | 68 | 19 (27.94) | 1.21 (0.70–2.10); 0.50 | 1.31 (0.72–2.40); 0.37 |

[a] Multivariable logistic regression model adjusted for age, gender, time to get drinking water, time to get washing water, latrine classification, and cluster;

[b] Five children lived in households with access that did not fit into the improved or unimproved category; bold denotes p<0.05.

positivity was more common in those with TF than those without (54.9% vs 28.3%; see Table 2 and Fig 3) but relatively stable by age (prevalence range: 30–41%) [S1 Table and Fig 2]). PCR positivity was associated with both TF (OR 2.65 [95% CI 1.71–3.55]) and seropositivity (OR 5.36 [95% CI 3.84–7.46]). Neither age (S1 Table; aOR 1.02 [95% CI 0.96–1.10]) nor gender (S1 Table; aOR 0.90 [95% CI 0.66–1.23]) affected the odds of infection.

## Anti-Pgp3 serology

Serological data were missing for 3.2% of participants; 5 refused, 8 did not have a DBS collected, and labelling errors preventing matching of DBS cards and clinical data for 13 participants. No data are available on reason for refusal. DBS were available from 792/818 (96.8%) children aged 1–9 years (Fig 1). Overall, seroprevalence by ELISA was 34.8% (276/792); the age-adjusted prevalence was 32.1% (95% CI 28.4%–36.3%). For children with TF, 52.3% (101/193) were seropositive compared with 29.2% (175/599) without TF (Table 2). Seropositivity increased from 9.0% in those aged 1 year to 48% in those aged 9 years (Fig 2). The estimated

**Table 2. Comparison of detection of ocular *Chlamydia trachomatis* by polymerase chain reaction and anti-Pgp3 antibodies by enzyme-linked immunosorbent assay in 1–9-year-old children with and without trachomatous inflammation—follicular in either eye, Nauru, July 2019.**

|  |  | PCR | | ELISA | |
|---|---|---|---|---|---|
|  |  | Total | Positive | Total | Positive |
| Clinical examination findings | TF | 193 | 106 (54.9%) | 193 | 101 (52.3%) |
|  | No TF | 587 | 166 (28.3%) | 599 | 175 (29.2%) |

Abbreviations: PCR, polymerase chain reaction; ELISA, enzyme-linked immunosorbent assay; TF, trachomatous inflammation—follicular.

seroconversion rate was 4.4% (95% CI 3.2–5.5%) per year of age, and the adjusted odds of being seropositive was 1.23-fold for each additional year of age in the 1–9-year-old range (S2 Table; aOR 1.23 [95% CI 1.15–1.32]). Seropositivity was associated with TF (OR 1.84 [95% CI 1.27–2.65]). The intensity of the ELISA response by year of age is shown in S1 Fig. Gender did not affect the odds of seropositivity (S2 Table, aOR 1.09 [95% CI 0.85–1.40]).

## Water, sanitation, and hygiene access

There was no association between any WASH access variable and TF (Table 1). In multivariable analysis, when compared with access to an improved source of drinking water, access to an unimproved drinking source was associated with a decreased adjusted odds of PCR positivity (S1 Table; aOR 0.29 [95% CI 0.19–0.44]) and an increased adjusted odds of seropositivity (S2 Table; aOR 1.70 [95% CI 1.02–2.86]). Access to a functioning handwashing facility that lacked soap was associated with an increased adjusted odds of seropositivity (S2 Table; aOR

**Fig 3. Co-occurrence of clinical and laboratory findings in children aged 1–9 years in Nauru, July 2019.**
Abbreviations: TF, trachomatous inflammation—follicular; ELISA, enzyme-linked immunosorbent assay; PCR, polymerase chain reaction.

2.49 [95% CI 1.00–6.19]). No other WASH variable was found to be significant in the multi-variable analysis.

## Discussion

In the first national survey to simultaneously assess clinical, bacteriological, serological, and WASH markers of trachoma in children aged 1–9 years in Nauru, we found high levels of all three biological indicators defining transmission intensity of ocular *C. trachomatis*: PCR, serology, and TF. We also found strong correlations among these indicators. Amongst 1–9-year-olds, the prevalence of TF and serological positivity increased with age, while the prevalence of infection remained stable. Taken together, and in contrast to the situation in several neighbouring Melanesian nations, this study provides evidence that children in Nauru are heavily affected by active trachoma and that the cause is infection with *C. trachomatis*.

The prevalence of ocular *C. trachomatis* infection in Nauruan children at 34.9% is the highest reported in the Western Pacific Region, and its relationship with TF is consistent with those reported in Kiribati (the one Pacific country confirmed to date to have a high prevalence of ocular *C. trachomatis* infection) and trachoma-endemic areas of Sub-Saharan Africa [15,28,29]. In Kiribati, 81%–90.7% of PCR-positive children were seropositive [15,28]. By contrast, in Nauru, we found only 61.4% to be seropositive. While there are individuals who are PCR positive without TF or seropositivity for chlamydial antibodies, generally these outcomes co-occur within individuals. PCR positivity in the absence of TF or seropositivity was more common in younger children, possibly reflecting individuals with early infections who did not yet have inflammatory sequalae resulting in clinical signs and without an exposure level sufficient for seroconversion.

According to current WHO guidelines, the prevalence of TF (21.8% [95% CI 15.2–26.2%]) in Nauruan children constitutes a public health problem requiring implementation of the A, F and E components of the SAFE strategy. The trichiasis prevalence in adults >15 years of age exceeded the 0.2% threshold defined as that determining whether chronic complications of trachoma are a public health problem, but none of the individuals diagnosed with trichiasis had TS identified. TT is the result of progressive conjunctival scarring and in trachoma-endemic settings, most individuals with trichiasis present with moderate-severe conjunctival scarring [30]. Trichiasis without TS may be due to age-related changes, prior trauma, or inflammatory conditions [30]. Therefore, the absence of scarring in any of the cases suggests the trichiasis reported here may not be trachoma-related. This finding, in the context of highly prevalent TF in children, presents a perplexing picture of the public health threat posed by trachoma in Nauru. Furthermore, children had a low prevalence of TI, a manifestation of severe active trachoma which, if present on repeated occasions over prolonged periods of observation, is the best known predictor of TS in adults [31].

There are several possible explanations for this pattern of results. Trachoma may have been only recently introduced to children in Nauru and intense *C. trachomatis* transmission has not been occurring for sufficiently long to produce cicatricial disease in adults. However, the 2007 trachoma rapid assessment found that a high proportion of TF (20.7%–33.0%) in 1–9-year-old children [9]. If high ocular *C. trachomatis* transmission is a recent local phenomenon, the decline in national living standards over the past few decades, coincident with the decline in income from phosphate mining in Nauru, might be responsible. Although trichiasis was found, the definition of trichiasis includes recent epilation, which may be done for reasons other than corneal abrasion [30]. Another possibility is that some of the TF, *C. trachomatis*, and antibody positivity may be due to inclusion conjunctivitis, caused by ocular infection with urogenital strains of *C. trachomatis*. Inclusion conjunctivitis can have the same clinical

manifestations as active trachoma and urogenital *C. trachomatis* has historically been reported to be highly prevalent in Nauru [32,33]. A recent report reported >20% of individuals tested were C. trachomatis positive. Further research investigating the contribution of specific *C. trachomatis* serovars to eye infection in Nauru via detailed molecular epidemiology of both ocular and genital *C. trachomatis* is required.

Access to sanitation has been associated with lower levels of both TF and *C. trachomatis* infection [34]. In comparison to neighbouring countries, Nauruan children have high levels of access to improved latrines (88.9%) [13,14,35]. This high level of sanitation coverage may explain the lack of association between latrine access and laboratory markers of chlamydial infection in this population. Despite the generally-held importance of household-level water access to trachoma elimination, the evidence from published research is inconsistent [34]. Almost all children in our study lived in households with access to improved sources of water for drinking (96.2%) and washing (95.7%), which may explain the lack of association of water variables and TF. Lack of soap at a functioning handwashing facility and an unimproved source of drinking water (not washing water) were associated with increased odds of anti-Pgp3 positivity, consistent with appropriate access to water and hygiene being beneficial. One unanticipated result was that an unimproved source of drinking water (not washing water) was associated with decreased odds of PCR positivity. It is possible that this result may be a type II error—finding an association where none exists or it may be due to an unmeasured confounding variable, such as socioeconomic status.

A number of limitations must be acknowledged. Firstly, PCR data on infection were missing for 4.6% of participants and serology missing for 3.2%. Second, as Pgp3 antibodies do not differentiate exposure to ocular from genital strains of chlamydia, seropositivity in an individual could reflect a range of potential exposures including trachoma, inclusion conjunctivitis, and perinatal transmission during childbirth [36,37]. An additional limitation shared by all standardised prevalence surveys using the Tropical Data method is that graders are trained to assess TS as "easily visible scarring" as per the WHO simplified grading system definition [22]. This encompasses moderate to severe scarring, and it is possible some mild scarring may be missed [38].

Finally, WASH variables collected as part of the Tropical Data methods were developed for settings where there is poor access to sanitation and water. In settings such as Nauru, where there is widespread access to safe household sanitation and water, the collection of additional variables associated with trachoma such as household crowding, and the frequency of face and hand washing should be considered.

As we approach global elimination more multi-modal, integrated screening and research, like this work, will be required. Given this, it would be ideal to have standardised methods for such work covering the capture, reporting, and interpretation of clinical, infection, and sero-survey data. Standardisation would allow for comparability across locations and provide a layer of quality control for reported results. Without such standardization, the assessment of scarring is difficult to interpret, and it is possible that cases of trichiasis are due to trachoma.

We set out to investigate the epidemiology of trachoma in Nauru. Our findings demonstrate the added value of using laboratory testing to provide richer insight. Clinical findings relating to trachoma across age-groups were not as expected, with prevalent childhood disease found in the absence of trichiasis plus TS in adults. While further work is required to understand this age-related discordance, our results show trachoma is rife in Nauruan children and suggest there has been a recent and substantial introduction of ocular *C. trachomatis*. The absence of TS and TT in adults further supports our conclusions. Based on these findings, Nauru implemented the SAFE strategy, including azithromycin mass drug administration for elimination of trachoma as a public health problem. Our approach of integrating collection of

clinical, laboratory, and WASH data, and assessing all four variables in parallel, can maximise understanding of the current state of trachoma epidemiology in new or unusual settings. Similar integrated public health work and research will be required to drive progress towards the global elimination of trachoma as a public health problem.

## Supporting information

**S1 Fig. Absorbance for ELISA by year of age.** Each dot represents an individual participant. Red horizontal line indicates the cut-off for positivity.
(TIF)

**S1 Table. Factors associated with *Chlamydia trachomatis* positive polymerase chain reaction (PCR) in children aged 1–9 years, Nauru, July 2019 (n = 780).**
(DOCX)

**S2 Table. Factors associated with Chlamydia trachomatis positive polymerase chain reaction (PCR) in children aged 1–9 years, Nauru, July 2019 (n = 780).**
(DOCX)

**S1 Text. Methods. Serological testing for anti-Pgp3 antibodies: Enzyme-linked immunosorbent assay.**
(DOC)

## Acknowledgments

We thank the residents of Nauru who gave their time to take part in this study. We also thank the field staff for their role in collecting data for the study. We thank Shanyko Benjamin for her role in laboratory support in Nauru. We thank Sarah Boyd of the International Trachoma Initiative, Cristina Jimenez of Sightsavers, Oliver Sokana of the Ministry of Health and Medical Services, Solomon Islands, and Sarity Dodson, Lizzie Jenkins, and Richard Le Mesurier from The Fred Hollows Foundation for their support of the study.

The authors alone are responsible for the views expressed in this article and they do not necessarily represent the views, decisions, or policies of the institutions with which they are affiliated.

## Author Contributions

**Conceptualization:** Kathleen D. Lynch, Sue Chen Apadinuwe, Stephen B. Lambert, Tessa Hillgrove, Sara Webster, Robert Butcher, Philip Cunningham, Diana Martin, Anthony W. Solomon, John M. Kaldor, Susana Vaz Nery.

**Data curation:** Kathleen D. Lynch, Robert S. Ware, Ana Bakhtiari.

**Formal analysis:** Kathleen D. Lynch, Stephen B. Lambert, Robert S. Ware, Ana Bakhtiari.

**Funding acquisition:** Sara Webster.

**Investigation:** Sue Chen Apadinuwe, Mitchell Starr, Beth Catlett, Chandalene Garabwan, Susana Vaz Nery.

**Methodology:** Kathleen D. Lynch, Stephen B. Lambert, Mitchell Starr, Beth Catlett, Emma M. Harding-Esch, Ana Bakhtiari, Philip Cunningham, Diana Martin, Sarah Gwyn, Anthony W. Solomon, John M. Kaldor.

**Project administration:** Kathleen D. Lynch, Stephen B. Lambert, John M. Kaldor, Susana Vaz Nery.

**Resources:** Sue Chen Apadinuwe, Philip Cunningham, Diana Martin, Sarah Gwyn.

**Software:** Ana Bakhtiari.

**Supervision:** Sue Chen Apadinuwe, Stephen B. Lambert, Tessa Hillgrove, Philip Cunningham, Diana Martin, Anthony W. Solomon, Chandalene Garabwan, John M. Kaldor, Susana Vaz Nery.

**Validation:** Mitchell Starr, Beth Catlett, Diana Martin, Sarah Gwyn.

**Visualization:** Kathleen D. Lynch, Stephen B. Lambert, Robert S. Ware, Sarah Gwyn.

**Writing – original draft:** Kathleen D. Lynch, Mitchell Starr, Beth Catlett, Robert S. Ware, Anthony W. Solomon.

**Writing – review & editing:** Kathleen D. Lynch, Sue Chen Apadinuwe, Stephen B. Lambert, Tessa Hillgrove, Mitchell Starr, Beth Catlett, Robert S. Ware, Anasaini Cama, Sara Webster, Emma M. Harding-Esch, Ana Bakhtiari, Robert Butcher, Philip Cunningham, Diana Martin, Sarah Gwyn, Anthony W. Solomon, Chandalene Garabwan, John M. Kaldor, Susana Vaz Nery.

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
