## [Decision Letter · Decision Letter 0]

1 Oct 2021

Dear Kathleen Lynch,

Thank you very much for submitting your manuscript "A national survey integrating clinical, laboratory, and WASH data to determine the typology of trachoma in Nauru" for consideration at PLOS Neglected Tropical Diseases. As with all papers reviewed by the journal, your manuscript was reviewed by members of the editorial board and by several independent reviewers. In light of the reviews (below this email), we would like to invite the resubmission of a significantly-revised version that takes into account the reviewers' comments. 

We cannot make any decision about publication until we have seen the revised manuscript and your response to the reviewers' comments. Your revised manuscript is also likely to be sent to reviewers for further evaluation.

Sincerely,

Scott D Nash

Guest Editor

Michael Marks

Deputy Editor

This manuscript describes a detailed trachoma survey in Nauru. A few minor comments:

-The LFA assay is first noted in the Data analysis section, and then later in the Results. Please include information on this assay either in the methods or in the supplemental material.

-The prevalence of TI is first mentioned in the Discussion. Please include this point in the Results section to help the reader.

-Please make sure C. trachomatis is italicized appropriately throughout the manuscript, including the abstract.

Reviewer's Responses to Questions

**Key Review Criteria Required for Acceptance?**

**Methods**

-Are the objectives of the study clearly articulated with a clear testable hypothesis stated?

-Is the study design appropriate to address the stated objectives?

-Is the population clearly described and appropriate for the hypothesis being tested?

-Is the sample size sufficient to ensure adequate power to address the hypothesis being tested?

-Were correct statistical analysis used to support conclusions?

-Are there concerns about ethical or regulatory requirements being met?

Reviewer #1: This was no doubt a well planned study and expensive to undertake. I don't understand why the WHO Simplified grading was used instead of a finer grading of active trachoma. If one is going to spend the time, the effort and the money to do PCR and serology, one would expect to at least grade trachoma with some precision. The Simplified grading scheme was designed for health workers to assess the public health impact of trachoma. To determine if the SAFE strategy or even MDA was needed in Nauru on could always also categorise the findings from a more specific gradings into TF or TT. It is highly likely that many children will have had F1 or F2 but not sufficient large follicles to be TF and equally many will have had C1 or C2 without reaching the threshold for TS!! 

For a very detailed research study like this, one wonders why were photos of the tarsal plate not taken?

Again if the challenge was to try to infer something about the history or trachoma in Nauru more detail should be provided about the findings in each of the age groups over 9 years of age.

Although WASH data were collected for the households, no data are presented for facial cleanliness. This is in many ways the key indicator for personal hygiene. it is very surprising that this was left out.

When a case was found the child was given 2 tubes of tetracycline ointment. Why wasn't the household treated with azithromycin or was MDA done later?

Reviewer #2: (No Response)

Reviewer #3: Generally, the methods are described adequately, with the exception of methods to detect and manage contamination in the field and laboratory. These need to be expanded as described below

while the sample size was done to provide estimates of TF, since further analyses were done with WASH variables, the authors should provide calculations of power to detect differences when rates of access to some variables were so high

**Results**

-Does the analysis presented match the analysis plan?

-Are the results clearly and completely presented?

-Are the figures (Tables, Images) of sufficient quality for clarity?

Reviewer #1: The data for adults should be presented appropriately broken down by age and gender.

The finding of one child under 10 with trichiasis is most unusual. This should be further explained as should the one case in those 10 to 15.

Reviewer #2: (No Response)

Reviewer #3: Table S1 should really be a table in the text itself

**Conclusions**

-Are the conclusions supported by the data presented?

-Are the limitations of analysis clearly described?

-Do the authors discuss how these data can be helpful to advance our understanding of the topic under study?

-Is public health relevance addressed?

Reviewer #1: The rates of TF are low in the children less that 5 years, although the PCR rates are high. Is this because these children had active trachoma but less that the 5 follicles 0.5mm needed to be called TF?

With the suggestion the "TS may not have been graded accurately" and the lack of a more detailed grading of tarsal scarring or photo documentation, these findings are really just speculation.

The "trachoma paradox" in the Pacific is a somewhat fanciful construct. It does not take into consideration that the rates and severity of trachoma can change over time. With population growth and increased crowding, without commensurate improvements in personal and community hygiene, trachoma can become more common and more severe.

The term "paratrachoma" was used by some for a while in the 1960s, as a term for chlamydial conjunctive infection that after a period of time resolved on its own without blinding sequelae. It seems this term has been misinterpreted. Paratrachoma was another name for "inclusion conjunctivitis". It could be caused by a single infection with any serovar. It was commonly seen with the genital strains because because there was usually only a single infective episode. The person was not subjected to the repeated episodes of reinfection required fro the development of "trachoma". The suggestion that these children, 24% of them, all had an almost simultaneous single infection seems just wrong.

Reviewer #2: (No Response)

Reviewer #3: The limitations in the discussion need to be expanded, and the conclusion around infection need to be re thought if there is in fact evidence of contamination.

**Editorial and Data Presentation Modifications?**

Reviewer #1: Data for the adults needs to be presented more fully.

If photos were taken of clean face data collected they need to be presented.

Reviewer #2: (No Response)

Reviewer #3: There needs to be much more clarity on how the potential for contamination of samples was addressed in the field and in the lab. This is a major concern. 

The authors also have to use TT in place of trichasis in several places, and by the results, they have to decide if they believe the TS data and what they found is not TT , or they don't and it is TT. The ramifications of either decision deserve further discussion.

**Summary and General Comments**

Reviewer #1: This is an experienced group, working on a small island, using some very sophisticated tests. However, they seem to have omitted the key elements required for the sort of study they are trying to present. If the the study were just a rapid assessment of trachoma, the PCR and serology are not needed. If this were an attempt to study the dynamics and kinetics of trachoma in Nauru, then they must use a finer grading system for trachoma and also present the data for the adults. If this were an attempt to look at the reasons trachoma might occur in Nauru, then in addition to the WASH factors, they must include facial cleanliness.

Reviewer #2: This is a well written an interesting study evaluating active trachoma, C. trachomatis, and serologic markers of C. trachomatis in Nauru. These data are interesting given the ongoing investigation of the “Pacific enigma”. 

1. Were WASH facilities observed or data collected by participant report only? (These may appear in the cited paper but a sentence so that readers don’t have to refer to the cited paper would be helpful).

2. How many household WASH indicators were entered into stepwise models and were these variables possibly collinear? Stepwise regression models can be a little problematic in that they tend to overfit, with over-estimated effect sizes and under-estimated variances. Depending on how many candidate variables are included (which isn’t clear from the methods), could all of the candidate variables be included (assuming there aren’t major problems with collinearity)? Looking at the supplemental tables, it’s a little unclear if a stepwise procedure was actually used (looks like all candidate variables are in the models), so if no stepwise procedure was used I would update the methods accordingly.

3. Page 20, Line 347 – could the authors speculate as to why they’re observing a lower seropositivity among PCR positives in their study versus Kiribati? 

4. Are there plans for antibiotic-based intervention for trachoma in Nauru (or, do the authors believe its warranted given these results)?

5. The relationship between WASH and trachoma variables could very well be due to unmeasured confounding, for example, by SES (in previous studies and in this one). It would be worth mentioning that cross-sectional associations shouldn’t infer causality and that there’s high risk of unmeasured confounding here. I think further survey work won’t really help us understand these questions, but randomized controlled trials might.

Reviewer #3: The manuscript adds data on trachoma in the Pacific Islands, and supports a more recent introduction of trachoma. My main concern is the possible presence of contamination in the infection results. Control measures in the filed or the lab are not described, and the results are strongly suggestive of a problem. That said, there needs to be infection in order to have contamination, so the likelihood that the infection data supports the finding of TF is strong. Nevertheless, this concern is given scant attention and needs to be addressed as described below 

Abstract: the conclusion that “the absence of trichiasis with trachomatous scarring suggest a relatively recent increase in transmission intensity” is liable to be confusing to readers who are unfamiliar with the distinction between trachomatous trichiasis (TT) and trichiasis due to other causes, especially as the abstract presented results on the prevalence of trichiasis as though it was TT, and thus were given importance like the rest of the trachoma related data. Either the authors believe their TS data and this is likely not trachomatous trichiasis, or they do not and the prevalence can be reported as TT. The sentence does need to be modified. Perhaps if that sentence on prevalence of trichiasis were deleted from the abstract, and a statement that although trichiasis was found, the absence of scarring in any of the cases suggested it may not be due to trachoma. Then the conclusion would make more sense to the reader, if that is what the authors think. 

Abstract: The Abstract needs to reflect the high rate of access to water and sanitation, then state no relationship was found with the presence of TF, so the reader quickly has an understanding of why that might be. 

Line 89-90:Introduction: Again, the description of the pathogenesis of trichiasis is really for trachomatous trichiasis and should be labeled as such, not just trichiasis. Here would be an excellent place to indicate that while there are other causes of trichiasis, TT is the result of progressive trachomatous scarring. 

Line 96: please change to “where prevalence of trachomatous trichiasis (TT) is >0.2% 

Line 124: I believe that one of the surveys in Peru collected all the indicators as well-if the authors mean the first national survey, because the entire nation was done in one survey, then they are likely right as very few countries are that small. 

Line 194: Given that the infection rate is so high, higher than the rate of TF, and does not go up with age, suggests that there may have been field and/or lab contamination. Can the control procedures to avoid field contamination be described? Were control swabs taken to check for field contamination? The high rate of infection positivity in those without disease is further concerning. 

Line 223: how many plates were not acceptable. Were the samples re run? 

Results and Figure 3. The data provide further supportive evidence of contamination. When fully 61% of positive test of infection is not accompanied by presence of disease, and half of those don’t even have positive serology, then one must investigate if contamination is an issue. We expect no disease but positive serology, given the cumulative nature of seropositivity-but that does not hold for infection

Line 389. Discussion: The authors need to consider that the results reflect a problem with the infection data-if indeed there was contamination, and several infection negative children were misclassified, this could produce the result found. 

Line 398: the limitation section needs more serious thought. How about the absence of data on other factors that may reflect transmission environment, like household crowding? Other WASH indicators may have found an association like frequency of face or handwashing-hence the need to put table S1 in the paper and not in the supplemental tables. Although mentioned briefly, the high proportion of children with very good access to water and improved latrines does limit the ability to find any association, and the authors could provide power calculations to back that up. As for scarring, are there any images of the eyelids in adults that could be checked for presence of scarring. Scarring severe enough to produce TT is usually easily visible, so that could be backed up with images. 

I find that Table S1 is more valuable than table 1, which can be easily summarized in the text, if only a few tables/figures are allowed. Table S1 provides more detail on the WASH variables, and also shows the problem of trying to find associations when there is scant dispersion.

PLOS authors have the option to publish the peer review history of their article (what does this mean?). If published, this will include your full peer review and any attached files.

Reviewer #1: No

Reviewer #2: No

Reviewer #3: No
---

## [Decision Letter · Decision Letter 1]

12 Jan 2022

Dear Kathleen Lynch,

Thank you very much for submitting your manuscript "A national survey integrating clinical, laboratory, and WASH data to determine the typology of trachoma in Nauru" for consideration at PLOS Neglected Tropical Diseases. As with all papers reviewed by the journal, your manuscript was reviewed by members of the editorial board and by several independent reviewers. The reviewers appreciated the attention to an important topic. Based on the reviews, we are likely to accept this manuscript for publication, providing that you modify the manuscript according to the review recommendations. 

Thank you to the authors for their thoughtful responses to the Reviewers' first round of comments. After further review, there remain 2 main issues that are concerning the reviewers. 

For the first point, about the possibility of contamination in the samples, the authors have been very clear in their explanation of contamination control during the laboratory work. For the possibility of field contamination, are the authors able to provide any more information as to how the samples were collected in the field, for example the training involved, and whether or not grader worked alone to do the work or whether there was assistance, possibly a "tube holder" if you will? The authors are recommended to either include the point in the limitation section as the reviewer suggests, or further evaluate/discuss the presence of high infection across the age groups, even among those without clinical signs or serological responses and what that may mean in this population.

For the second point on the grading of TS, while they may not have "standardized" their TS grading, the authors are clear in their responses that they trust their graders. To increase trust among readers, the authors need to further detail the Tropical Data training that they used In Nauru as it relates to training on TT and TS specifically, including what classroom and/or field training went into the assessment of TT and TS. The authors are further welcome to recommend whether they think enhanced training and/or data collection would have been useful.

Sincerely,

Scott D Nash

Guest Editor

Michael Marks

Deputy Editor

Thank you to the authors for their thoughtful responses to the Reviewers' first round of comments. After further review, there remain 2 main issues that are concerning the reviewers. 

For the first point, about the possibility of contamination in the samples, the authors have been very clear in their explanation of contamination control during the laboratory work. For the possibility of field contamination, are the authors able to provide any more information as to how the samples were collected in the field, for example the training involved, and whether or not grader worked alone to do the work or whether there was assistance, possibly a "tube holder" if you will? The authors are recommended to either include the point in the limitation section as the reviewer suggests, or further evaluate/discuss the presence of high infection across the age groups, even among those without clinical signs for or serological responses and what that may mean in this population.

For the second point on the grading of TS, while they may not have "standardized" their TS grading, the authors are clear in their responses that they trust their graders. To increase trust among readers, the authors need to further detail the Tropical Data training that they used as it relates to training on TT and TS specifically, including what classroom and/or field training went into the assessment of TT and TS. The authors are further welcome to recommend whether they think enhanced training and/or data collection would have been useful.

Reviewer's Responses to Questions

**Key Review Criteria Required for Acceptance?**

**Methods**

-Are the objectives of the study clearly articulated with a clear testable hypothesis stated?

-Is the study design appropriate to address the stated objectives?

-Is the population clearly described and appropriate for the hypothesis being tested?

-Is the sample size sufficient to ensure adequate power to address the hypothesis being tested?

-Were correct statistical analysis used to support conclusions?

-Are there concerns about ethical or regulatory requirements being met?

Reviewer #2: (No Response)

Reviewer #3: the authors have responded to reviewer concerns

**Results**

-Does the analysis presented match the analysis plan?

-Are the results clearly and completely presented?

-Are the figures (Tables, Images) of sufficient quality for clarity?

Reviewer #2: (No Response)

Reviewer #3: the autbors have responded to reviewer concerns

**Conclusions**

-Are the conclusions supported by the data presented?

-Are the limitations of analysis clearly described?

-Do the authors discuss how these data can be helpful to advance our understanding of the topic under study?

-Is public health relevance addressed?

Reviewer #2: (No Response)

Reviewer #3: The authors have gone to great lengths to assure us of the absence of contamination in acquisition of the ocular swab and in laboratory processing. They have also referenced the one article by Solomon et al, where very high rates of PCR positivity were seen in three settings , in some cases greater than the rates of TF. this article was published in 2003 before the necessity of careful standardization in both laboratory and in the field was appreciated. While the collection of air swabs is not a perfect way to guarantee absence of contamination, it is one way to provide some assurances there was none. To cite detailed protocols is helpful but without some data to provide reassurances of absence of contamination, the possibility must be entertained, if at least in limitations. it is difficult to see data where there is no age effect in infection, yet an age effect in seropositivity, and in disease. I believe the authors cannot exclude the possibility of field contamination in their data, despite the protocol, and that mention of this possibility must be present in the limitations. With all we have learned in the 19 years years since the cited study, it is difficult to believe that so many young children had infection without either disease or without being seropositive. 

Secondly, the absence of standardization of scarring in grading is a concern when it is to be used to decide if trichiasis is due to trachoma. The authors have decided these cases are not due to trachoma. The limitations state that the graders may have missed mild scarring but without standardization, we do not know if they could have missed moderate scarring. The limitations should say that without standardization, the assessment of scarring is difficult to interpret and it is possible these are due to trachoma. The authors have gone to great lengths to assure us of the absence of contamination in acquisition of the ocular swab and in laboratory processing. They have also referenced the one article by Solomon et al, where very high rates of PCR positivity were seen in three settings , in some cases greater than the rates of TF. this article was published in 2003 before the necessity of careful standardization in both laboratory and in the field was appreciated. While the collection of air swabs is not a perfect way to guarantee absence of contamination, it is one way to provide some assurances there was none. To cite detailed protocols is helpful but without some data to provide reassurances of absence of contamination, the possibility must be entertained, if at least in limitations. it is difficult to see data where there is no age effect in infection, yet an age effect in seropositivity, and in disease. I believe the authors cannot exclude the possibility of field contamination in their data, despite the protocol, and that mention of this possibility must be present in the limitations. With all we have learned in the 19 years years since the cited study, it is difficult to believe that so many young children had infection without either disease or without being seropositive. 

Secondly, the absence of standardization of scarring in grading is a concern when it is to be used to decide if trichiasis is due to trachoma. The authors have decided these cases are not due to trachoma. The limitations state that the graders may have missed mild scarring but without standardization, we do not know if they could have missed moderate scarring. The limitations should say that without standardization, the assessment of scarring is difficult to interpret and it is possible these are due to trachoma.

**Editorial and Data Presentation Modifications?**

Reviewer #2: (No Response)

Reviewer #3: (No Response)

**Summary and General Comments**

Reviewer #2: (No Response)

Reviewer #3: (No Response)

PLOS authors have the option to publish the peer review history of their article (what does this mean?). If published, this will include your full peer review and any attached files.

Reviewer #2: No

Reviewer #3: No

Figure Files:

Data Requirements:

Reproducibility:

References

---

## [Editor Report · Decision Letter 2]

24 Feb 2022

Dear Ms. Lynch,

We are pleased to inform you that your manuscript 'A national survey integrating clinical, laboratory, and WASH data to determine the typology of trachoma in Nauru' has been provisionally accepted for publication in PLOS Neglected Tropical Diseases.

Best regards,

Scott D Nash

Guest Editor

Michael Marks

Deputy Editor

---

## [Editor Report · Acceptance letter]

4 Apr 2022

Dear Ms Lynch,

We are delighted to inform you that your manuscript, "A national survey integrating clinical, laboratory, and WASH data to determine the typology of trachoma in Nauru," has been formally accepted for publication in PLOS Neglected Tropical Diseases.

Best regards,

Shaden Kamhawi

co-Editor-in-Chief

Paul Brindley

co-Editor-in-Chief
